# Validation of an ultra-short global quality of life scale in a large population-based health survey

John-Kåre Vederhus[1]*, Christine Timko[2], Bente Birkeland[3], Siri Håvås Haugland[3], Karin Berle Gabrielsen[1]

1 Addiction Unit, Sørlandet Hospital, Kristiansand, Norway, 2 Center for Innovation to Implementation, Veterans Affairs Health Care System and Stanford University School of Medicine, Palo Alto, CA, United States of America, 3 Department of Psychosocial Health, University of Agder, Grimstad, Norway

* john-kare.vederhus@sshf.no

**Data Availability Statement:** The data file is available from the SIKT - Surveybanken database

## Abstract

### Introduction

Quality of life (QoL) assessment is essential in health-related research and clinical settings, offering insights into individuals' well-being and functioning. This study validated the Essential QoL-3 (EQoL-3), an ultra-short scale assessing essential dimensions of QoL, for use in epidemiological research and clinical settings.

### Methods

Data from a 2021 national survey in Norway (N = 17,487) were used. Three items on the EQoL-3 assess life satisfaction, happiness, and meaningfulness on a 0–10 scale. Discriminant validity was assessed by comparing the EQoL-3 with the Satisfaction with Life Scale (SWLS) and by examining latent mean differences between individuals with adverse life experiences (ALE+) and those without such experiences (ALE-). Convergent validity was evaluated through latent regression analyses comparing the EQoL-3 with a perceived mastery scale and a mental distress scale.

### Results

The discriminant validity of the EQoL-3 was less than optimal when compared with the SWLS. Nonetheless, a multigroup confirmatory factor analysis revealed that the EQoL-3 score was 1.42 (95% CI = 1.33–1.50, p < 0.001) lower in the ALE+ group compared to the ALE- group, providing support for discriminant validity. Convergent validity was established with a positive association between EQoL-3 and mastery (β = 1.75, 95% CI = 1.70–1.80, p < 0.001) and a negative association between the EQoL-3 and mental distress (β = -2.64, 95% CI = -2.71/-2.59, p < 0.001).

### Conclusions

The EQoL-3 is a reliable measure of QoL. Its streamlined nature facilitates quick administration, making it a valuable tool for clinicians and researchers in diverse settings. Its inclusion

(accession number NSD-NSD2995-V1). https://doi.org/10.18712/NSD-NSD2995-V1.

**Funding:** The author(s) received no specific funding for this work.

of the eudaimonic dimension, as well as its exclusion of health items in the measure itself, distinguishes it from traditional HQoL measures, making it suitable for mental health and substance use disorder research.

## Introduction

The assessment of quality of life (QoL) has long been a crucial component of health-related research [1]. It provides valuable insights into individuals' subjective experiences, functioning, and well-being, and contributes to evaluating the impact of interventions and healthcare outcomes [2, 3]. QoL assessment also plays a pivotal role in epidemiological research, providing valuable insights into the disease burden across a diversity of populations worldwide [4].

There is a wide range of QoL measures that incorporate the evaluation of health within their constructs, commonly known as health-related QoL (HQoL) measures [5, 6]. However, existing measures often involve lengthy and comprehensive assessments that may be burdensome in certain contexts. Even the condensed version of WHO's QoL measure (WHOQoL-BREF) has 26 items [7]. While there are shorter HQoL measures such as the SF-12 and the RAND-12, they primarily focus on the impact of health issues on other aspects of life such as functioning, social conditions, and on overall QoL [6, 8]. A typical wording is: "Does your health now limit you in these activities? If so, how much [9]?" Consequently, these measures imply that overall QoL is mainly a function of health-related factors, i.e., they have adopted a "health first" attitude.

A common use of QoL data is to benchmark patient data against data from reference populations without the condition in question, allowing researchers to assess the impact of the patient's condition on individuals [5]. However, typical HQoL measures that focus on the aspects of QoL affected by health may introduce bias when comparing these populations, since health issues may not hold the same level of importance or be as essential to individuals without existing health problems. Similarly, in the field of mental health and addiction, there are concerns regarding the use of HQoL measures that emphasize physical health and physical functioning, such as the EQ-5D [10, 11]. Many studies in this field have found that mental health conditions and substance use disorder (SUD) have a negative impact on physical health [12, 13]. Yet, physical health factors are not essential to the condition itself. Moreover, it is worth noting that individuals being assessed may interpret the term "health" primarily in relation to physical well-being [14]. This limited interpretation may lead to the often-devastating impacts of mental health or SUD conditions being overlooked or not fully understood when individuals are asked to rate their "health".

In the mental health and SUD field, it will be more suitable to utilize HQoL measures that allow respondents to evaluate health independently from its consequences on functioning. For instance, the QoL-5, consisting of five items, has been successfully used in several studies in the field of addiction research [15, 16]. Such measures would consider physical and mental health as just one aspect of QoL alongside other dimensions, such as quality of relationships. An example is the question regarding physical health in the QoL-5: «How would you rate your physical health at the moment." However, the inclusion of an item in the QoL-5 where individuals rate their perceived mental health status introduces a difficulty. Two studies within this field examining the association between relevant factors and QoL, measured with the QoL-5, found that mental distress, measured with the Hopkins Symptom Checklist-10 item version, was the strongest factor explaining variations in QoL [16, 17]. While mental health negatively

impacts QoL, there is a question of whether this association is influenced by circular reasoning since an indicator of mental health is included in the QoL measure itself. Thus, the inclusion of health-related items in a QoL measure that are similar or overlapping with the health measure used as a predictor can make it challenging to distinguish the unique contribution of health as a predictor of overall QoL. Therefore, particularly within the mental health and SUD research field, there is a need for brief yet reliable assessment tools that exclude health-related items from the measure itself.

In 2018, the Norwegian Institute of Public Health (NIPH), in collaboration with Statistics Norway (SSB), conducted a review of QoL measures used in European public health surveys, and based on OECD guidelines for measuring subjective well-being, they recommended that QoL surveys in Norway should include three dimensions to provide a comprehensive assessment [18, 19]. These dimensions are the cognitive evaluation dimension, which is assessed through a satisfaction with life item; the affective dimension, which captures emotional well-being and can be measured using items related to happiness or other relevant emotions; and the eudaimonic dimension, which assesses the sense of purpose and meaningfulness in life [18]. The cognitive and affective items correspond to the concept of hedonia, which is deeply rooted in the tradition of subjective well-being (SWB) and is widely recognized as a fundamental aspect of an individual's quality of life [20, 21]. Eudaimonic well-being encompasses an individual's sense of life's purpose and has received increased attention in QoL research over the past decade [22, 23].

Related to the affective dimension, the NIPH/SSB suggested a rather lengthy list of positive and negative emotions (e.g., feeling happy, calm and relaxed, worried, lonely, anxious) [18]. However, including a single positive item, such as happiness, should be sufficient to assess this dimension, as respondents can rate their emotional state on a scale from worst to most positive. This argument also aligns with psychometric theory when applying scales in a latent variable framework. In such a framework, questions function as reflective indicators representing their respective constructs. In a reflective model, the items are in principle interchangeable, ensuring consistent interpretation of the model regardless of the number of indicators, as long as the included indicators are highly relevant and provide sufficient identification for the model [24].

To address the need for a concise yet robust measurement tool, our study presents an ultra-short QoL measure that condenses the essential dimensions of QoL into three concise items, incorporating both the hedonic and eudaimonic dimension of QoL [22]. Although this measure has been previously employed [25, 26], it has not been formally validated, warranting a comprehensive evaluation to establish its internal consistency, validity, and utility.

## Objective

The objective of the present study is to validate this ultra-short QoL measure, called Essential QoL-3 (EQoL-3). As a part of this process, we examined the scale's discriminant and convergent validity. To assess discriminant validity, we compared the EQoL-3 with a traditional SWB scale, which focuses specifically on global life satisfaction as a single dimension of QoL. We also examined whether respondents who had adverse life experiences (ALEs) in the previous year would differ from those without such experiences. We hypothesized that the group with ALE would have significantly lower QoL [27, 28]. To assess convergent validity, we hypothesized a positive correlation between the EQoL-3 and the sense of being in control of one's life, also known as perceived mastery [29–31]. Additionally, we expected a negative association between the EQoL-3 and a scale assessing mental distress, in alignment with previous studies [16]. This would indicate the usefulness of the scale in a clinical setting.

## Materials and methods

### Participants and procedures

The present investigation draws upon data obtained from a national survey executed by SSB in the year 2021, commissioned by the Norwegian Directorate of Health [32]. The primary aim of this survey was to assess QoL across diverse demographic segments. The data collection process involved a normative sample of 40,000 residents aged 18 years or older, selected at random from the Norwegian population. These individuals were approached via email, postal mail, or SMS, and were provided with comprehensive information pertaining to the survey, alongside a web-based survey link. The total number of responses received was 17,487, equating to a response rate of 44%. Data collection took place in the spring of 2021.

### Measures

Demographic information, including age, gender, educational attainment and employment status, was obtained from the National Population Registry after the completion of the survey. Respondents also reported their living arrangements, i.e., whether they lived alone. They were also asked whether they had encountered significant ALEs in the past year, such as separation/divorce, serious economic problems, serious illness, having a child with severe disability, experienced violence or threat of violence against oneself, experienced being forced or pressured to sexual intercourse, being degraded, or entering into unemployment. The presence of any of these experiences within the last year was used as a grouping variable, resulting in an ALE + group and an ALE- group.

### Quality of life

As outlined above, the EQoL-3 scale has three items, assessing perceived satisfaction with life, perceived happiness, and perceived meaningfulness in life (Table 1). Thus, the indicators of the EQoL-3 were considered to represent a univariate construct, namely the essential features of QoL. Participants rated the items on a scale of 0 to 10, with higher scores indicating greater satisfaction, happiness, and perceived meaningfulness. In this study, we employed latent construct analysis following standard convention in Structural Equation Modeling (SEM). Unlike the traditional formative approach of averaging the scores of the three items on the scale, we adopted a more robust method. Using SEM procedures, we estimated the mean of the averaged latent construct (QoL) by considering the variance/covariance structure between items, which in SEM terminology is called indicators. This method accounts for measurement errors associated with each observed indicator. In essence, it assesses how effectively each indicator

**Table 1.  The questions of the Essential QoL-3 scale (EQoL-3) with English translation.**

| Item | Questions: |
|------|------------|
| **Q1** | Alt i alt, hvor fornøyd er du med livet ditt for tiden? |
|  | **Translation:** Overall, how satisfied are you with your life right now? |
| **Q2** | Alt i alt, i hvilken grad opplever du at det du gjør i livet er meningsfylt? |
|  | **Translation:** Overall, to what extent do you feel the things you do in your life are meaningful? |
| **Q3** | Tenk på hvordan du har følt deg de siste 7 dagene. I hvilken grad var du glad? |
|  | **Translation:** Think about the past 7 days. How happy did you feel? |

Note: Participants rated the items on a scale of 0 to 10, with higher scores indicating greater satisfaction, perceived meaningfulness, and happiness.

represents the underlying latent construct and provides a more accurate representation of the construct.

## Life satisfaction

The Satisfaction with Life Scale (SWLS) was developed by Diener et al. and is widely recognized as one of the most used instruments for measuring SWB [21, 33]. The developers explicitly stated that the scale does not encompass the affective aspect of SWB; rather, it was designed solely to assess life satisfaction. Furthermore, it was developed as a global measure, aiming to evaluate individuals' contentment with their lives as a whole, rather than specific areas such as health or finances. The scale has demonstrated favorable psychometric properties, exhibiting high levels of internal consistency and test-retest reliability [21]. Comprising five items, such as "I am satisfied with my life" and "The conditions of my life are excellent," the scale offers seven response options ranging from disagreement to agreement. Higher scores correspond with greater levels of satisfaction with life.

## Mental distress

A condensed version of the Hopkins Symptom Checklist called the SCL-5 was employed to assess mental distress [34]. Individuals were asked to indicate whether they had felt sad or depressed, hopeless about the future, tense or keyed up, constantly fearful and anxious, or worried during the previous week. Responses were scored on a 4-point ordinal scale ranging from "not bothered" to "extremely bothered." The SCL-5 has shown comparable effectiveness to more extensive variations like the SCL-10 and SCL-25, and its reliability was excellent according to a validation study in Norway (Cronbach's alpha = 0.87) [35].

## Mastery

The Sense of Mastery Scale assesses an individual's perception of personal control over their life chances rather than being ruled by fate [29]. It includes 5 negatively phrased statements such as "There is little I can do to change many of the important things in my life" and "I often feel helpless in dealing with the problems of life." Participants rate these statements on a 5-point Likert scale ranging from 1 - "Strongly disagree" to 5 - "Strongly agree." In this study, the scores were reversed, such that higher scores indicated higher levels of mastery.

## Ethics approval and consent to participate

The online survey began with an introductory page that included information about the study's purpose and a consent form, which respondents had to click on and agree to before they were directed to the questionnaire. Thus, the provision of informed consent in the survey was contingent on participation in the survey (implied consent). The survey endeavors conducted by SSB conform to the statutory requisites delineated in the Statistics Act and align with the principles set forth in the General Data Protection Regulation of the European Union [36]. Subsequently, following a one-year interval, the acquired data was subjected to a process of anonymization and deposited within the Norwegian Agency for Shared Services in Education and Research (SIKT) data repository, where it is readily accessible for scientific purposes [37]. The need for ethical approval in this study was waived by the Regional Committee for Medical Research Ethics, region South-East, as the data were fully anonymized before being transferred to the SIKT data repository (REK no. 712698).

## Statistical analyses

Descriptive statistics were used to show sample characteristics. To establish the discriminant validity of the EQoL-3, we employed the multitrait-multimethod matrix to calculate the heterotrait-monotrait ratio (HTMT) of correlations between two constructs [38]. This method examines the correlations between indicators within the same construct (monotraits) and the correlations between indicators of different constructs (heterotraits). A HTMT ratio of less than 0.85 is considered to be a conservative threshold, indicating that the measures being compared are sufficiently distinct from each other [38]. In our study, we examined the correlation between the EQoL-3 and the SWLS. Given that the SWLS was not designed to capture the affective and eudaimonic aspects of well-being included in the EQoL-3, our hypothesis was that the HTMT value would be less than 0.85. To calculate the HTMT value, we utilized Henseler's online matrix [39]. We evaluated the internal consistency of the EQoL-3 with the Cronbach's alpha coefficient [40]. An alpha value greater than 0.70 is generally considered acceptable.

In order to further confirm the discriminant validity of the EQoL-3, we employed a second method called multigroup confirmatory factor analysis (CFA) to determine if the EQoL-3 differentiated between groups. This analysis was conducted within a SEM framework [41]. Before comparing means, we established the equivalence (invariance) of the scale across groups, i.e., that the indicators were understood similarly in both groups [42]. First, a baseline model was established for each group, presented with the standardized factor loadings (β). Then, a simultaneous multigroup CFA was conducted, progressively imposing greater constraints on measurement parameters to test for strong measurement invariance, i.e., scalar equivalence, between the groups. Scalar equivalence implies that the scales have the same factor structure (configural equivalence), equivalent factor loadings (metric equivalence), and similar intercepts across groups [43]. Measurement equivalence was assessed by comparing the nested models using chi-square differences ($\Delta\chi2$) in relation to the change in degrees of freedom ($\Delta$df) [44]. A non-significant $\Delta\chi2$ value indicates that the constraints do not significantly worsen the model fit, supporting the null hypothesis of measurement invariance [45]. When scalar equivalence is achieved, it is appropriate to compare latent means. Latent mean differences are reported as unstandardized beta (β) values, along with 95% confidence intervals (CIs).

To establish the convergent validity of the EQoL-3 measure, we investigated its association with mastery and mental distress through a latent regression analysis [24, p. 212]. The results were reported using unstandardized β coefficients to facilitate interpretation. Additionally, the R squared value ($R^2$) was calculated to estimate the proportion of variation in EQoL-3 that could be explained by the model.

Mplus version 8.6 was employed to conduct the analyses. The maximum likelihood estimator (ML) was utilized [44]. The default procedure in Mplus, full information maximum likelihood, handled missing values. The assumptions for the analyses were met, including a sufficient sample size for the CFA. A significance level of $p < 0.05$ was adopted. Model fit was evaluated using the Root Mean Square Error of Approximation (RMSEA) and the Comparative Fit Index (CFI). An RMSEA of $\leq 0.08$ indicated an acceptable fit and $\leq 0.06$ a good model fit. A CFI $\geq 0.95$ indicated a good model fit [46, 47].

## Results

The data set had minimal missing values, with coverage exceeding 99.7% for all variables. The final sample size for the analysis was 17,477, after excluding 10 cases with missing values for the group variable. The survey achieved a representative sample at the national level, ensuring

**Table 2. Sample characteristics (N = 17,487).**

| Variables | N (%) or mean (±) |
|---|---|
| Age | 49 (±17) |
| Sex (female %)[a] | 9,071 (52) |
| Employed / working [a] | 12,571 (72) |
| Educational level (N = 17,087) [a] | |
| Primary and secondary school (up to 10 years of education) | 2,556 (15) |
| High school (up to 13 years of education) | 6,616 (39) |
| University college or university (≥ bachelor's degree) | 7,915 (46) |
| Nationality and immigrant status | |
| Native Norwegians | 15,250 (87) |
| Immigrants | 2,075 (12) |
| Born in Norway of immigrant parents | 162 (1) |
| Living arrangement (N = 17,472) | |
| Married or registered partnership | 8,470 (48) |
| Cohabitation (not registered) | 3,618 (21) |
| Other arrangements (e.g., living with children, living alone) | 5,384 (31) |
| At least one adverse life event in the last 12 months[b] | 3,891 (22) |

[a] Retrieved from the National population registry

[b] Experienced at least one of these adverse life events in the last year: separation/divorce, serious economic problems, serious illness, had a child with severe disability, experience violence or threat of violence against oneself, experienced being forced or pressured to sexual intercourse, being degraded or becoming unemployed.

that the characteristics of the entire population of adults aged 18 and above in Norway were closely reflected. About one-half of participants were female (52%), and the average age of the sample was 49 years, with a standard deviation of 17 years (Table 2). Almost half of participants had a bachelor's degree or higher. The majority (87%) were native Norwegians and roughly 2 out of 3 were living in a partnership. Moreover, 22% of the sample reported having faced an ALE within the year leading up to the survey (Table 2).

## Internal consistency and discriminant validity of EQoL-3

The formative mean of the averaged scale for the sample as a whole was 6.86 (SD 1.97). The scale had good internal consistency, as evidenced by a Cronbach's alpha value of 0.86. The multitrait-multimethod matrix between the EQoL-3 and the SWLS is shown in Table 3. The HTMT ratio between the scales was calculated to be 0.88, indicating that they are not distinct measures when marker of 0.85 is used.

Therefore, we conducted a secondary analysis using multigroup CFA. Before conducting the CFA analysis, we assessed the measurement model of the EQoL-3 in both groups using a one-factor CFA solution. As expected, the model displayed a "perfect" fit, with an RMSEA of 0.00 and a CFI of 1.00, as a model with only three indicators is "just-identified" and goodness-of-fit criteria do not apply. Yet, in terms of factor loading estimates, the suggested solution exhibited a consistent factor structure across both groups, with factor loadings being significant below the 0.001 level, and no measurement errors identified. The standardized factor loadings ranged from 0.74 to 0.91 in the ALE+ group and from 0.70 to 0.89 in the ALE- group.

The multigroup CFA specifying the $\chi2$ and CFI values for the configural, metric, and scalar models are shown in Table 4. The $\Delta\chi2$s between the nested models were not significant, suggesting that the measure was perceived as equivalent across groups (Table 5). The scalar model

**Table 3. Multitrait-multimethod matrix of the EQoL-3 and the satisfaction with life scale.**

|  | Satis-faction | Meaning-fulness | Happi-ness | LS1 | LS2 | LS3 | LS4 | LS5 |
|---|---|---|---|---|---|---|---|---|
| Satisfaction | 1.00 |  |  |  |  |  |  |  |
| Meaningfulness | 0.75 | 1.00 |  |  |  |  |  |  |
| Happiness | 0.66 | 0.60 | 1.00 |  |  |  |  |  |
| LS1 | 0.69 | 0.64 | 0.58 | 1.00 |  |  |  |  |
| LS2 | 0.60 | 0.55 | 0.51 | 0.71 | 1.00 |  |  |  |
| LS3 | 0.74 | 0.68 | 0.63 | 0.81 | 0.74 | 1.00 |  |  |
| LS4 | 0.55 | 0.55 | 0.49 | 0.65 | 0.59 | 0.67 | 1.00 |  |
| LS5 | 0.50 | 0.48 | 0.45 | 0.58 | 0.51 | 0.58 | 0.61 | 1.00 |

The green and yellow fields represent within measure correlations and the red fields represent between measure correlations. Abbreviations: Essential Quality of Life 3 scale = EQoL-3, LS1 –LS5 = Indicators of the Satisfaction with Life Scale.

also had excellent fit statistics (Table 4). As a result, the latent means could be compared. The EQoL-3 score was 1.42 (95% CI = 1.33–1.50, p<0.001) lower in the ALE+ versus the group without such experiences.

## Convergent validity

The initial latent regression model examining the relationship between QoL and perceived mastery showed inadequate model fit, with an RMSEA greater than 0.09. MPlus's modification indices highlighted measurement errors in the mastery measure. This error correlation has also been mentioned in a validation study of this scale [48]. When we adjusted for this error correlation, the model had an acceptable model fit, with an RMSEA of 0.07 and CFI of 0.98. The model revealed a significant and strong positive association between the measures (β = 1.75, 95% CI = 1.70–1.80, p < 0.001, Fig 1), explaining 57% of the variance in QoL. In contrast, there was a strong negative association between the mental distress scale and EQoL-3 (β = -2.64, 95% CI = -2.71/-2.59, p < 0.001, Fig 2), and it explained 62% of the variance in QoL. Again, some error correlations in the mental distress measure had to be adjusted. After the adjustments, the model had good fit statistics with RMSEA = 0.06 and CFI = 0.99.

## Discussion

This study established the EQoL-3 scale as a reliable and valid tool for assessing essential and global QoL. Analyses provided evidence of the scale's robust psychometric properties, including its discriminant validity. Specifically, individuals who had ALEs reported significantly lower QoL compared to those who had not encountered such experiences. Moreover, the

**Table 4. Multigroup confirmatory factor analysis results of the measurement invariance test across the two groups: ALE+ (N = 3,891) and ALE- (N = 13,586)[a].**

|  | $\chi^2$ | *Df* | RMSEA | CFI |
|---|---|---|---|---|
| Configural model | 0.00 | 0 | 0.00 | 1.00 |
| Metric model | 0.01 | 2 | 0.00 | 1.00 |
| Scalar model | 4.75 | 4 | 0.01 | 1.00 |

[a] N = 10 had missing on the group variable used in the analysis. Abbreviations: ALE = Adverse Life Events, Df = degrees of Freedom, CFI = comparative fit index, RMSEA = root mean square error of approximation.

**Table 5. Multigroup confirmatory factor analysis results of the measurement invariance test across the two groups: ALE+ (N = 3,891) and ALE- (N = 13,586)[a] – Δχ2 between nested models.**

|  | Δχ2 | Df | P-value for Δχ2 |
|---|---|---|---|
| Metric against configural model | 0.00 | 2 | 0.957 |
| Scalar against configural model | 0.01 | 4 | 0.314 |
| Scalar model against metric model | 4.75 | 2 | 0.097 |

[a] N = 10 had missing on the group variable used in the analysis. Abbreviations: Δχ2 = change in χ2 between models, Df = degrees of Freedom.

EQoL-3 scale demonstrated convergent validity as it exhibited expected associations with perceived mastery.

The successful validation of a measure that captures the essential dimensions of QoL underscores its potential value in clinical and epidemiological research. Notably, the EQ3 includes the eudaimonic dimension of QoL, which assesses the individual's perception of meaningfulness in life—a vital element reflecting the potential for personal growth and fulfillment [49]. Eudaimonic well-being taps into deeper sources of satisfaction and fulfillment that extend beyond immediate pleasure [22, 23]. The quote by Victor Frankl, "The person who has a why to live for can bear with almost any how," suggests that this dimension can even surpass the importance of the hedonic aspects of QoL [50, p. 72].

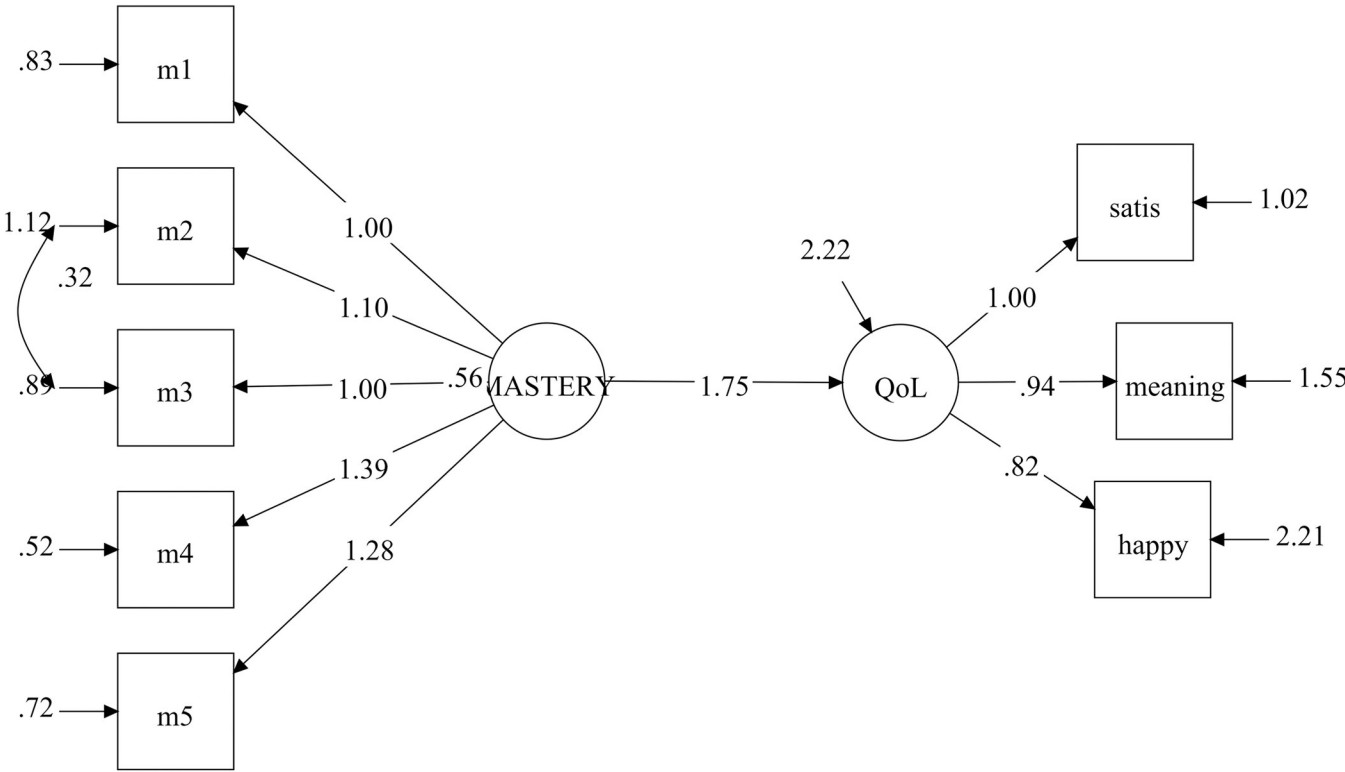

**Fig 1. Latent regression analysis showing the association between perceived mastery[a] and QoL measured with the Essential QoL-3 (EQoL-3) scale.** [a] Measured with the Sense of Mastery Scale. The figure shows the measurement and the structural model with unstandardized factor loadings. Abbreviations: QoL = quality of life, m1-m5 = indicators of the Sense of Mastery Scale, satis = satisfaction, meaning = meaningfulness, happy = happiness.

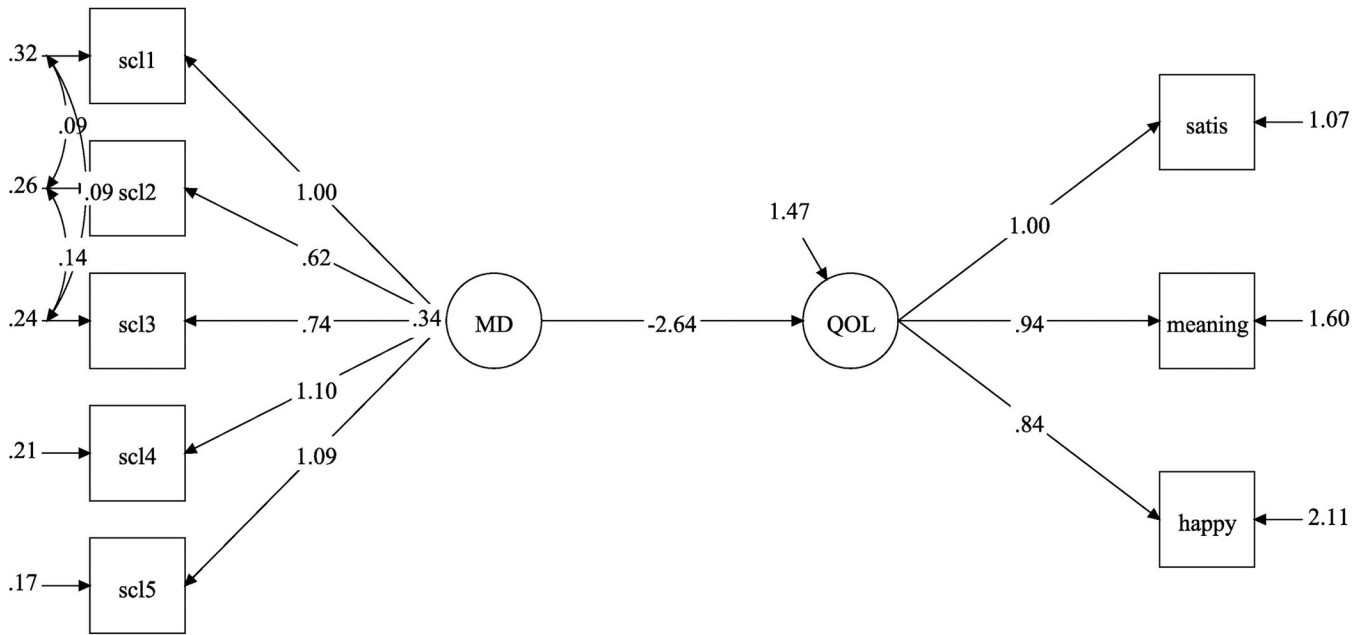

**Fig 2. Latent regression analysis showing the association between mental distress [a] and QoL measured with the Essential QoL-3 (EQoL-3) scale.** [a] Measured with the Hopkin Symptom Check List– 5 item version (SCL-5). The figure shows the measurement and the structural model with unstandardized factor loadings. Abbreviations: MD = mental distress, QoL = quality of life, scl1-scl5 = indicators of the SCL-5, satis = satisfaction, meaning = meaningfulness, happy = happiness.

There are several advantages to distilling QoL to its core elements, rather than including constructs that have the ability to influence QoL (such as health) as inherent aspects of the construct itself. This approach allows for the identification of predictors that influence QoL, providing a more comprehensive understanding of its determinants. This is especially relevant in the fields of mental health and SUD research. By using global QoL tools that do not include health as an inherent aspect, concerns related to commonly used HQoL measures like the EQ-5D can be addressed, particularly within these fields [10]. This choice also avoids circular reasoning and allows for a more accurate assessment of the impact that health challenges have on individuals. Individuals without health concerns may not prioritize their importance as much, but when health is threatened, it will likely become a more salient issue. About 40 years ago, Diener et al. emphasized the importance of assessing individuals' overall evaluation of life, rather than simply aggregating satisfaction ratings across different life domains, such as health, work, and social relationships [21]. Their approach acknowledges that individuals may assign different values to various aspects of their lives at different times and emphasized the value of global QoL measures. Every person can relate in some way to the essential concepts included in the EQoL-3, regardless of their health status. This statement is also supported by our finding of the equivalence of the scale across groups. Thus, by using global QoL measures without the health dimension included, it is more appropriate to compare patient data with non-patient reference populations.

Unexpectedly, when applying the HTMT procedure to evaluate the discriminant validity of the EQoL-3 by comparing it with the SWLS, the result was not convincing. This surprising finding implies that the SWLS, initially believed to solely capture the dimension of life satisfaction within an essential QoL framework, may exhibit more overlap with the EQoL-3 than anticipated. Some of the SWLS items seem to capture whether individuals are living in

alignment with their perceived life purpose, such as "In most ways, my life is close to my ideal" and "If I could live my life over, I would change almost nothing." This suggests that the SWLS may encompass elements related to eudaimonic well-being despite its original intent to solely assess life satisfaction.

Nonetheless, as the multigroup CFA revealed that the ALE+ group had a significantly lower score on the EQoL-3 compared to individuals without such experiences, our findings support the discriminant validity of the EQoL-3. The observed latent mean difference between the two groups was 1.42, which is considered substantial on a 0–10-point scale. These findings are consistent with previous studies that reported similar results, highlighting the detrimental impact of ALEs on QoL [27, 28].

The latent regression analysis found a significant and positive connection between the EQoL-3 and mastery. This provided support for the EQoL-3's convergent validity, meaning it is highly correlated in the expected direction with a related construct, i.e., the perceived sense of control over challenges and life stressors [51]. According to standard interpretation methods, an increase of one unit in the mastery scale was associated with a notable 1.75-point increase in the EQoL-3 score. Furthermore, in line with expectations, there was a negative association between the EQoL-3 and the mental distress scale, i.e., that higher levels of mental distress were associated with lower levels of QoL. The magnitude of this association was even stronger, albeit in the opposite direction, with a 2.64-point decrease in the EQoL-3 score for each unit increase in the mental distress scale. These findings provide further support for the clinical utility of the EQoL-3 scale, indicating that it can effectively differentiate between individuals with varying levels of mental distress. These results are consistent with previous studies that have also found a positive correlation between an increased sense of control and mastery in life and higher levels of QoL, as well as studies that have discovered a negative correlation between QoL and mental distress [16, 30, 31, 51].

## Methodological considerations

Strengths of the study include its use of a large, nationally representative population survey, which allows for generalizability of findings, making them applicable for informed policy decisions and interventions. Including participants across key characteristics such as age, gender and socioeconomic status enables meaningful comparisons across different subgroups, as demonstrated in our study comparing the ALE+ and ALE- groups. The SEM methodology provides a more accurate and precise representation of the latent construct and strengthens the validity of our findings. In addition to strengths of the study methods, the EQoL-3 itself has important strengths. Recognizing that lengthy QoL questionnaires are not feasible in larger-scale surveys, the streamlined nature of the EQoL-3 provides an advantageous solution. By enabling quick administration and data collection, the EQoL-3 minimizes participant burden and retains the assessment's informative value. Thus, the utilization of this scale is expected to be highly advantageous for both epidemiological research and clinical settings. Its compact nature facilitates seamless integration into routine patient assessments, enabling easy monitoring of treatment progress. Additionally, the user-friendly design makes it a convenient outcome measure for clinical use. Regarding limitations, it should be kept in mind that our findings were based on cross-sectional data. This means that the findings cannot determine, for example, whether a perceived sense of control and mastery in life causes higher QoL. Indeed, it is likely that there is a bi-directional association between these factors. Test-retest reliability was not assessed. Nevertheless, the current validation results provide strong evidence for the reliability and validity of the measure. Future research incorporating test-retest reliability would further reinforce these findings.

## Conclusions

The validation of the EQoL-3 scale as a comprehensive and feasible-to-use measure suggests that it has the ability to provide clinicians and researchers with valuable insights into individuals' overall QoL. By evaluating QoL holistically, this scale allows for a better understanding of the factors that influence QoL, including the impact of complex conditions, such as mental health and SUDs.

## Acknowledgments

We acknowledge the Statistics Norway and the SIKT for making the data publicly available. There was no external funding of this study. CT was supported by the U.S. Department of Veterans Affairs (HSR&D RCS 00–001).

## Author Contributions

**Conceptualization:** John-Kåre Vederhus.

**Data curation:** John-Kåre Vederhus.

**Formal analysis:** John-Kåre Vederhus.

**Validation:** Christine Timko, Bente Birkeland, Siri Håvås Haugland, Karin Berle Gabrielsen.

**Writing – original draft:** John-Kåre Vederhus, Karin Berle Gabrielsen.

**Writing – review & editing:** Christine Timko, Bente Birkeland, Siri Håvås Haugland.

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
