## [Decision Letter · Decision Letter 0]

7 Jun 2024

PONE-D-24-11217Validation of an ultra-short global Quality of Life scale in a large population-based health surveyPLOS ONE

Dear Dr. Vederhus,

Thank you for submitting your manuscript to PLOS ONE. After careful consideration, we feel that it has merit but does not fully meet PLOS ONE’s publication criteria as it currently stands. Therefore, we invite you to submit a revised version of the manuscript that addresses the points raised during the review process.

We look forward to receiving your revised manuscript.

Kind regards,

Angelina Wilson Fadiji, PhD

Academic Editor

PLOS ONE

Journal Requirements:

Additional Editor Comments:

Dear Authors,

Please consider the following reviewer comments in reviisong your manuscript.

Reviewer comments

Overall, the article presents a well-structured and clear validation study of the Essential QoL-3 (EQoL-3) scale. The introduction effectively sets the stage for the need for a concise yet robust QoL measure, particularly in mental health and addiction research. The literature review is comprehensive, covering the importance of QoL assessment, limitations of existing measures, and the theoretical framework underlying the EQoL-3.

The methods section is detailed and transparent, providing sufficient information to replicate the study. The sample size is adequate, and the data analysis is appropriate for the research questions.

The results section is clearly presented, with tables and text that effectively convey the findings. The EQoL-3 demonstrates excellent internal consistency, discriminant validity, and convergent validity, supporting its use as a valid and useful measure of QoL.

The discussion section effectively summarizes the main findings, relates them to the broader literature, and discusses the implications of the study. The authors convincingly argue that the EQoL-3 offers a concise and robust measure of QoL that can be used in various contexts.

However, some minor suggestions for improvement include:

• Providing more context for the EQoL-3 scale, such as its development process and previous uses

• Including more information about the sample demographics and characteristics

• Discussing potential limitations of the study, such as the reliance on self-report measures

• Exploring future research directions and potential applications of the EQoL-3 in more detail

To elaborate on the question on whether the statistical analysis been performed appropriately and rigorously?

I wonder why only a multigroup CFA was conducted. In current practice, several measurement models are tested, for instance a bifactor model, an ESEM and even a bi-factor ESEM. I suggest the authors do a more robust analysis or get the support of a statistician to do this.

Also, while the author/s conducted convergent and discriminant validity, other types of reliability such as test-rest reliability need to be determined for a psychometric study.

The author/s did not acknowledge any assumptions made during their data analysis.

The following concerns below should be further addressed:

How did you ensure data quality?

Was the study pre-registered? If not, why not?

In your CFA, which identification strategy did you employ and why (unit variance identification, or unit loading identification)?

Which methods of setting factor loading did you use and why? Fixed factor loading or freely estimated factor loading?

Overall, the article presents a strong validation study of the EQoL-3 scale, and with some minor revisions, it has the potential to make a significant contribution to the field of QoL research.

Reviewers' comments:

Reviewer's Responses to Questions

**Comments to the Author**

1. Is the manuscript technically sound, and do the data support the conclusions?

Reviewer #1: Yes

Reviewer #2: Yes

2. Has the statistical analysis been performed appropriately and rigorously? 

Reviewer #1: No

Reviewer #2: Yes

3. Have the authors made all data underlying the findings in their manuscript fully available?

Reviewer #1: Yes

Reviewer #2: Yes

4. Is the manuscript presented in an intelligible fashion and written in standard English?

Reviewer #1: Yes

Reviewer #2: Yes

5. Review Comments to the Author

Reviewer #1: Ejoke Ufuoma concerns

Overall, the article presents a well-structured and clear validation study of the Essential QoL-3 (EQoL-3) scale. The introduction effectively sets the stage for the need for a concise yet robust QoL measure, particularly in mental health and addiction research. The literature review is comprehensive, covering the importance of QoL assessment, limitations of existing measures, and the theoretical framework underlying the EQoL-3.

The methods section is detailed and transparent, providing sufficient information to replicate the study. The sample size is adequate, and the data analysis is appropriate for the research questions.

The results section is clearly presented, with tables and text that effectively convey the findings. The EQoL-3 demonstrates excellent internal consistency, discriminant validity, and convergent validity, supporting its use as a valid and useful measure of QoL.

The discussion section effectively summarizes the main findings, relates them to the broader literature, and discusses the implications of the study. The authors convincingly argue that the EQoL-3 offers a concise and robust measure of QoL that can be used in various contexts.

However, some minor suggestions for improvement include:

• Providing more context for the EQoL-3 scale, such as its development process and previous uses

• Including more information about the sample demographics and characteristics

• Discussing potential limitations of the study, such as the reliance on self-report measures

• Exploring future research directions and potential applications of the EQoL-3 in more detail

To elaborate on the question on whether the statistical analysis been performed appropriately and rigorously?

I wonder why only a multigroup CFA was conducted. In current practice, several measurement models are tested, for instance a bifactor model, an ESEM and even a bi-factor ESEM. I suggest the authors do a more robust analysis or get the support of a statistician to do this.

Also, while the author/s conducted convergent and discriminant validity, other types of reliability such as test-rest reliability need to be determined for a psychometric study.

The author/s did not acknowledge any assumptions made during their data analysis.

The following concerns below should be further addressed:

How did you ensure data quality?

Was the study pre-registered? If not, why not?

In your CFA, which identification strategy did you employ and why (unit variance identification, or unit loading identification)?

Which methods of setting factor loading did you use and why? Fixed factor loading or freely estimated factor loading?

Overall, the article presents a strong validation study of the EQoL-3 scale, and with some minor revisions, it has the potential to make a significant contribution to the field of QoL research.

Reviewer #2: Dear authors,

Congratulations on the manuscript! I found it very interesting, especially because it is a simple and quick instrument for assessing QoL, very useful especially for use in large population studies. The inclusion of aspects of cognitive, affective and eudaimonic dimensions are very welcome, as most of the instruments used in research are related to health-related quality of life, which may not have much meaning or importance for individuals without an actual health problem health. The objectives and methodology of the work are also well described. In general, I have no further observations to make, other than to congratulate you once again.

6. PLOS authors have the option to publish the peer review history of their article (what does this mean?). If published, this will include your full peer review and any attached files.

Reviewer #1: **Yes: **Ejoke Ufuoma Patience PhD

Reviewer #2: No

---

## [Author Response · Author response to Decision Letter 0]

25 Jun 2024

See enclosed "Response to Reviewers" letter.

---

## [Editor Report · Decision Letter 1]

5 Jul 2024

Validation of an ultra-short global Quality of Life scale in a large population-based health survey

PONE-D-24-11217R1

Dear Dr. Vederhus,

We’re pleased to inform you that your manuscript has been judged scientifically suitable for publication and will be formally accepted for publication once it meets all outstanding technical requirements.

Kind regards,

Angelina Wilson Fadiji, PhD

Academic Editor

PLOS ONE
---

## [Editor Report · Acceptance letter]

21 Aug 2024

PONE-D-24-11217R1 

PLOS ONE

Dear Dr. Vederhus, 

I'm pleased to inform you that your manuscript has been deemed suitable for publication in PLOS ONE. Congratulations! Your manuscript is now being handed over to our production team.

Kind regards, 

on behalf of

Dr. Angelina Wilson Fadiji 

Academic Editor

PLOS ONE